# Novelty Unlocking with Multiobjective Generative Models: Intra-Batch Diversity of Human Motions

## Abstract

Current generative models have shown potential performance in many tasks, which typically focus on generating samples that closely adhere to a given distribution. However, existing methods cannot simultaneously generate diverse and high-quality samples, and they fail to produce optimal diverse solutions in *intra-batch diversity* . Recognizing that maintaining "diversity" has been a longstanding challenge in multiobjective optimization, we were inspired to introduce a multi-objective optimization approach to enhance diversity in a single pass. This paper utilizes the in-betweening human motion generation task as an example and introduces the multiobjective generative models to produce a batch of diverse and smooth human motion sequences in one pass. The resulting method, termed the *Multiobjective Generation Framework with In-Betweening Motion Model* (MGF-IMM), frames the human motion in-betweening task as a bi-objective optimization problem. The designed in-betweening motion model is then integrated into a multiobjective optimization framework to address this bi-objective optimization problem. Through comprehensive experiments, MGF-IMM has surpassed the state-of-the-art methods and validated its superiority in generating diverse in-betweening human motions, without introducing additional training parameters. Github

## 1 Introduction

Generative models Wei et al. (2024); Goodfellow et al. (2020); Ho et al. (2020); Shafir et al. (2024), such as Variational Autoencoders (VAEs) Kingma et al. (2019), Generative Adversarial Networks (GANs) Goodfellow et al. (2020), and Diffusion models Ho et al. (2020), have recently emerged as a promising technique in many tasks, such as computer animation Harvey et al. (2020), virtual reality Zhao et al. (2022), and human-computer interactions Mohammed et al. (2020). It aims to learn the underlying distribution of the data, thus supporting the generation of new samples. For example, in human motion generation fields, Wei *et al.* Mao et al. (2022) utilized a VAE-based approach combined with Transformer network to fill long-term missing motion frames. Zhou *et al.* Harvey et al. (2020) proposed a conditional GAN to learn in-betweening human motions. Tevet *et al.* Tevet et al. (2023) presented a human motion diffusion model for in-betweening human motion generation, they adapted a classifier-free diffusion-based generative model for this task.

Despite achieving success in many applications, generative models still face challenges. For instance, they struggle to consistently produce diverse and high-quality samples after a single training process. Additionally, generative models are highly sensitive to hyperparameters. On large datasets, setting training parameters appropriately is crucial, as retraining the model is resource-intensive and time-consuming. This means that current generative models cannot guarantee *intra-batch diversity* . It is important to note that the need for generating "multiple" and "diverse" outputs is quite common in real-world scenarios. For example, in in-betweening human motion generation task Tevet et al. (2023), the objective is to create a set of diverse and smooth sequences in one pass to interpolate user-provided sequences, offering engineers a wide range of transitional human motions, which has a wide range of applications in areas, such as AI for animation Xiao et al. (2024).

Coincidentally, maintaining "diversity" is also a long-standing issue in the domain of multiobjective optimization Guo et al. (2024); Song et al. (2014); Hua et al. (2021); Wang et al. (2014). In general, multiobjective optimization problems involve multiple conflicting objectives, which means that no single solution can simultaneously minimize or maximize all objectives Liu et al. (2022); Qiao et al. (2022). This impressive characteristic leads us to transform the generative task into a multiobjective optimization problem. Surprisingly, due to the implicit parallelism of the population-based search strategy, evolutionary algorithms (EAs) Tian et al. (2020a) can not only provide multiple Pareto optimal solutions in a single pass, but their well-designed diversity maintenance strategies also facilitate solutions that are well-distributed along the Pareto front. Thus EAs are utilized to optimize the designed multiobjective problem and generate diverse samples. In this work, we employ in-betweening human motion generation task as a showcase to report the effectiveness of the designed multiobjective generative models.

Currently, it is difficult to capture human motion data, as all motion types and variations require a tremendous amount of time and effort. Additionally, human motion exhibits a high degree of diversity, with significant individual differences. Therefore, efficiently generating diverse and smooth in-betweening human motion sequences remains a significant challenge. While the diversity of generative models for human motion has been discussed in recent years Guo et al. (2020), they often fail to guarantee that human motion sequences generated through batch sampling will differ significantly from one another. Intra-batch diversity is crucial for users, who are typically more concerned with the diversity of the sequences they receive. Therefore, this paper focuses on maintaining intra-batch diversity to address this need.

Motivated by this consideration, this paper designs an *Multiobjective Generation Framework with In-Betweening Motion Model* (MGF-IMM) for diverse in-betweening human motion task. We demonstrate that, without introducing any additional training parameters, the proposed MGF-IMM effectively guides the generation of diversity and smoothness in-betweening human motions. This is supported mainly by the following three techniques:

- *Multiobjective Modeling of Human Motion In-Betweening*: We transform the human motion in-betweening task into a multiobjective optimization problem. We prove that for any Pareto optimal solutions of the formulated problem, the corresponding human motion sequences can support smooth and diverse transitions to interpolate user-provided sequences. Moreover, we also demonstrate that two points on the Pareto front that are far apart, usually correspond to two motion sequences with different motion labels.

- *Multiobjective Generation Framework*: In the optimization process, a generative model is incorporated into the multiobjective EA to generate diverse human motion sequences. Specifically, the generative model is used to generate in-between motion frames between the user-provided motion sequence frames, and environmental selection is employed to guide the generated motions to evolve toward the Pareto optimal solutions.

- *In-Betweening Motion Model*: We developed a Transformer-based in-betweening motion model within our multiobjective generation framework to facilitate the generation of in-betweening motion sequences. While we primarily focus on the designed in-betweening motion model based on the VAE generative model, the designed architecture is also adaptable to other generative methods such as GAN and DDPM. Our results indicate that, regardless of whether the VAE, GAN, or DDPM architecture is employed, the designed multiobjective generation framework successfully guides diverse and smooth in-betweening human motion sequences.

## 2 RELATED WORK

### 2.1 MULTIOBJECTIVE OPTIMIZATION

Multiobjective optimization problems involve optimizing multiple conflicting objectives simultaneously. In general, the multiobjective optimization problem can be formulated as:

$$\min : \mathbf{f}(\mathbf{x}) = (f_1(\mathbf{x}), ..., f_M(\mathbf{x})) \tag{1}$$

where $\mathbf{x} = (x_1, \ldots, x_D) \in \mathcal{X} \subset \mathbb{R}^D$ is the decision vector (e.g., a candidate motion sequence), $x_i$ ($i \in \{1, ..., D\}$) is the $i$th decision variable (e.g., a component of a motion sequence), $D$ is the

dimension of the decision vector, $\mathcal{X}$ is the decision space, $\mathbf{f}(\mathbf{x}) \in \mathcal{Y} \subset \mathbb{R}^M$ is the objective vector, $f_i(\mathbf{x})(i \in \{1, \ldots, M\})$ is the $i$th objective, $M$ is the number of objectives, and $\mathcal{Y}$ is the objective space. Key concepts associated with the multiobjective optimization are as follows Deb (2011). *Pareto Dominance*: For decision vectors $\mathbf{x}_a$ and $\mathbf{x}_b$, if $\forall i \in \{1, 2, \ldots, m\}$, $f_i(\mathbf{x}_a) \leq f_i(\mathbf{x}_b)$ and $\exists j \in \{1, 2, \ldots, m\}$, $f_j(\mathbf{x}_a) < f_j(\mathbf{x}_b)$, $\mathbf{x}_a$ is said to Pareto dominate $\mathbf{x}_b$. *Pareto-Optimal Solution*: If no decision vector in $\mathbb{X}$ Pareto dominates $\mathbf{x}_a$, then $\mathbf{x}_a$ is a Pareto-optimal solution. *Pareto Set*: The set of all Pareto-optimal solutions forms the Pareto set in decision space. *Pareto Front*: The image of the Pareto set in the objective space forms the Pareto front.

In recent decades, researchers have developed various algorithms and frameworks to address multiobjective optimization problems Li et al. (2023a); Vodopija et al. (2024); Pan et al. (2024); Lee et al. (2018). EAs, due to their inherent parallelism and population-based search strategies, have become a preferred method for converging towards a set of optimal solutions within a single optimization run Branke (2008). Their impressive performance and straightforward implementation have led to widespread applications across real-world multiobjective optimization problems Tian et al. (2020b;a); Liu et al. (2021). One of the most appealing features of EAs is their ability to maintain population diversity. Such mechanisms are crucial for discovering solutions that are well-distributed along the Pareto front. Therefore, we are inspired to transform our task into a multiobjective optimization problem.

## 2.2 DIVERSE IN-BETWEENING HUMAN MOTION TASK

Diverse in-betweening human motion task aims to generate diverse and smooth motion sequences given user-provided sequences. A plethora of generative models, such as VAEs and GANs, have been applied to this task Mao et al. (2022); Fragkiadaki et al. (2015); Li et al. (2023b); Agrawal et al. (2013). For example, Harvey *et al.* Harvey et al. (2020) utilized neural networks to generate plausible interpolation human motions between a given pair of keyframe poses. Tang *et al.* Tang et al. (2022) built upon these findings by introducing a Convolutional Variational Autoencoder (CVAE) Mao et al. (2022), which utilizes the motion manifold and conditional transitioning to generate real-time motion transitions. Motion DNA Zhou et al. (2020) proposed to automatically synthesize complex motions over a long time interval given very sparse keyframes by users for long-term in-betweening. However, VAEs assume a Gaussian distribution as the posterior, which can limit the diversity of the generated samples. Meanwhile, GANs tend to mainly generate samples from the major modes while ignoring the minor modes, further restricting the overall diversity. More recently, denoising diffusion models Ho et al. (2020); Lee et al. (2023); Guo et al. (2020) have been extensively utilized for motion generation. MoFusion Dabral et al. (2023) proposed a denoising diffusion-based framework to generate motion in-betweening by fixing a set of keyframes in the motion sequence and reverse-diffusing the remaining frames. OmniControl Xie et al. (2024) presents a novel method for text-conditioned human motion generation, which incorporates flexible spatial control signals over different joints at different times. Their diffusion backbone is based on the human motion diffusion method. Based on this, CondMDI Cohan et al. (2024) proposes a flexible in-betweening pipeline through random sampling keyframes and concatenating binary masking during training. The above state-of-the-art methods handle diverse motion sequences by enhancing generative models to produce a wide range of plausible samples. However, they overlook the issue of ensuring intra-batch diversity in the sampling process. In contrast, this paper introduces a multiobjective sampling method to ensure high intra-batch diversity. Our approach requires no additional training and can be seamlessly integrated into various generative models as a plug-and-play solution.

## 3 MULTIOBJECTIVE MODELING OF THE HUMAN MOTION IN-BETWEENING TASK

In this section, we present the modeling of the diverse human motion in-betweening task as a multiobjective optimization problem. The formulated multiobjective optimization problem consists of bi-objective functions, which aim to control the diversity and smoothness of the generated motions, referred to as the *Diversity Component* and *Smoothness Component* respectively.

**Diversity Component**: We assume the availability of a classifier $C(Y)$ that can categorize the motion type represented by a generated motion sequence $Y$. We consider $D$ distinct types of motions, each labeled as $\{0, 1, ..., D - 1\}$. Thus, for any sequence $Y$, we have $C(Y) \in \{0, 1, ..., D - 1\}$. In

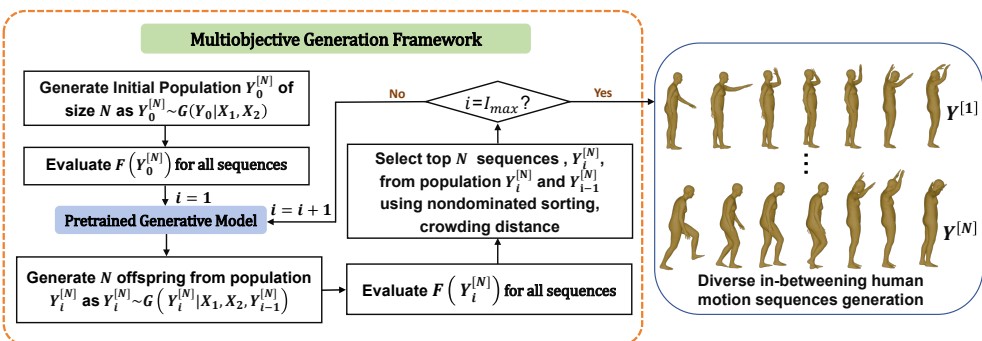

Figure 1: The illustration of multiobjective generation framework for diverse in-betweening human motion sequences. $Y$ is the generated sequence. $G$ denotes the pretrained generator of the generative model. $F$ denotes the designed multiobjective function.

addition, the classifier can provide the probability that a given motion sequence belongs to each categorized motion type. This probability is denoted as $P_c(Y)$. Based on the above definition, the diversity component is formulated as the following two functions $\alpha_1(Y)$ and $\alpha_2(Y)$:

$$\begin{cases} \alpha_1(Y) = \frac{1}{D} \left( C(Y) + P_c(Y) \right), \\ \alpha_2(Y) = 1 - \alpha_1(Y). \end{cases} \tag{2}$$

***Smoothness Component***: A smoothness function $\beta(Y)$ is utilized to facilitate the smoothness of adjacent motion sequences, which is formulated as follows:

$$\beta(Y) = ||X_1[-1] - Y[0]|| + ||Y[-1] - X_2[0]||, \tag{3}$$

where $X_1[-1]$ and $Y[-1]$ denote the last human pose of the user-provided motion sequence $X_1$ and the in-betweening sequence $Y$, respectively. $X_2[0]$ and $Y[0]$ denote the first human pose of the motion sequence $X_2$ and the generated in-betweening sequence $Y$, respectively. $\beta(Y)$ is utilized to denote the offset between the generated in-betweening motion sequences $Y$ and the adjacent motion sequences ($X_1$ and $X_2$). This operation aims to guarantee that the changes between the adjacent human motion poses are not too large, thereby enhancing the smoothness between different action sequences.

By integrating the diversity and smoothness components, the multiobjective optimization problem for the diverse human motion in-betweening task can be formulated as follows:

$$\begin{cases} \min : F_1(Y) = \alpha_1(Y) + \beta(Y), \\ \min : F_2(Y) = \alpha_2(Y) + \beta(Y). \end{cases} \tag{4}$$

Next, we introduce two theorems to formally summarize the key characteristics of the modeled multiobjective optimization problem.

**Theorem 1** *Let $\mathcal{B} = \{Y_b | Y_b = \arg\min_Y \beta(Y)\}$. Assuming $|\mathcal{B}| \geq 2$, we can state that any in-betweening motion $Y_b \in \mathcal{B}$ is a Pareto optimal solution of the multiobjective optimization problem defined in equation 4.*

The proof of *Theorem 1* can be found in the Appendix. *Theorem 1* establishes that all in-betweening human motion sequences that minimize the smoothness component $\beta(\cdot)$ are Pareto optimal solutions for the multiobjective optimization problem defined in equation 4. This result implies that solving equation 4 to identify its Pareto optimal motions inherently favors those with low, or even minimum, values of $\beta(\cdot)$. Given that $\beta(\cdot)$ governs the smoothness of transitions between an initial human pose and subsequent in-betweening motions, the Pareto optimal solutions are expected to facilitate smooth motion transitions, thereby ensuring that the generated motions exhibit continuity and smoothness.

**Theorem 2** *Let $\mathcal{B} = \{Y_b | Y_b = \arg\min_Y \beta(Y)\}$, and $Y_1, Y_2 \in \mathcal{B}$. If $||\boldsymbol{F}(Y_1), \boldsymbol{F}(Y_2)|| > \frac{4}{D}$, where $\boldsymbol{F}(\cdot) = (F_1(\cdot), F_2(\cdot))$ and $|| \cdot ||$ is the Manhattan distance, then $C(Y_1) \neq C(Y_2)$.*

The proof of *Theorem 2* can be found in the Appendix. *Theorem 2* demonstrates that, under the condition of the smooth transition, if the Manhattan distance between two generated in-betweening motions in the objective space exceeds $\frac{4}{D}$, the human motion sequences are likely to belong to different motion categories. This finding suggests that a diverse set of Pareto optimal motions from equation 4 not only maintains smooth transitions but also provides a variety of motion types. The multiobjective optimization framework is particularly suited to achieving this goal, offering a robust mechanism for generating smooth and diverse in-betweening motions.

## 4 MULTIOBJECTIVE GENERATION FRAMEWORK WITH IN-BETWEENING MOTION MODELS

### 4.1 MULTIOBJECTIVE GENERATION FRAMEWORK

EAs are highly effective for solving multiobjective optimization problems. It is logical to employ EAs to address the multiobjective optimization problem described in Section 4. However, it is crucial to recognize that human motion sequences are represented by a high-dimensional set of vectors, which introduces substantial challenges in the search for the Pareto optimal solutions. Furthermore, these sequences must adhere to constraints to ensure physically meaningful representations. These factors complicate the direct application of EAs for searching Pareto-optimal solutions within the human motion space. A viable solution to these challenges involves utilizing generative models in place of traditional EA operations, such as crossover and mutation, to produce offspring. Generative models can encode high-dimensional data into a low-dimensional space, which avoids directly generating offspring within the high-dimensional human motion space and ensures that the generated offspring comply with the necessary constraints Wong et al. (2023). Motivated by these considerations, we propose a multiobjective generation framework for human motion in-betweening task.

The designed multiobjective generation framework is illustrated in Figure. 1. In general, it follows the most common framework of genetic algorithms, including the following steps:

**Step 1.** *Initialization*: The pretrained generative model is employed to produce an initial population of human motion sequences. These sequences are subsequently evaluated using the objective functions defined in equation 4.

**Step 2.** *Offspring Generation*: Generating a set of offspring human motion sequences using the pretrained generative model.

**Step 3.** *Evaluation*: Evaluating the offspring using the objective functions defined in equation 4.

**Step 4.** *Environmental Selection*: Performing fast nondominated sorting and calculating the crowding distance Deb et al. (2002) for the sequences in both the current population and the offspring population. Subsequently, update the population through the elite selection.

**Step 5.** Repeating steps 2 to 4 until the termination condition is met.

The uniqueness of the proposed method lies in the incorporation of a generative model into the multiobjective optimization process. Specifically, the generation process is formulated as follows:

$$Y_i^{[n]} \sim \begin{cases} G\left(Y_i \mid X_1, X_2\right) & i = 0 \\ G\left(Y_i^{[n]} \mid Y_{i-1}^{[n]}, X_1, X_2\right) & 1 \leq i \leq I \end{cases}, \tag{5}$$

where $G$ is the generative model, $Y_i^{[n]}$ ($n \in \{1, ..., N\}$) is the $n$th human motion sequence in the population at the $i$th iteration. It can be observed that in the first iteration, i.e., during the initialization process, the human motion sequence is generated by the conditional generative model on the user-provided sequences $X_1$ and $X_2$. In subsequent iterations, i.e., during the evolution process, the generative model is conditioned not only by the user-provided sequences but also by the sequences already generated and present in the current population. This approach allows the generative model to search for Pareto-optimal sequences based on the elite sequences identified thus far, thereby driving the evolution of the population. After achieving the max iterations, we can get $N$ optimal solutions, i.e., diverse and smooth in-betweening human motion sequences.

Moreover, to allow for more flexible in-betweening human motion generation, the length of the generated motion sequence is variable, as different action sequences may require different transition

lengths. The desired motion length is estimated according to the following steps: 1) Calculating the cosine similarity $S \in [0, 1]$ to measure the similarity between two given motion poses $X_1[-1]$ and $X_2[0]$. 2) Based on the preset minimum sequence length $Y_{min}$ and the preset maximum sequence length $Y_{max}$, the desired length of the human motion sequence is set as follows:

$$Y_{len} = Y_{min} + \lfloor (Y_{max} - Y_{min}) \times (1 - S) \rfloor . \tag{6}$$

In our experiment, $Y_{min}$ is set to 5, $Y_{max}$ is set to 15. In the generation of the offspring, the $Y_{len}$ is encoded into the pretrained generative model and has to be consistent with the training process details. So we introduce a padding operation to the $Y_{len}$, specifically, we repeat the last pose of the $Y_{len}$ until it achieves the $Y_{max}$.

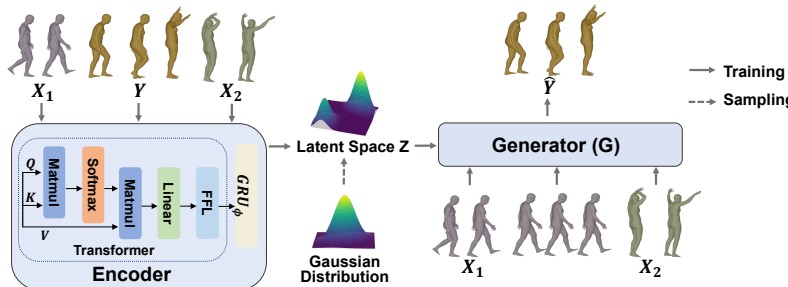

Figure 2: The illustration of pretrained generative model using VAE, which is utilized to generate human motion sequences in multiobjective generation framework.

## 4.2 IN-BETWEENING MOTION MODELS

To enhance the generation of human motion sequences, we have designed a in-betweening motion model. Notably, by simply altering the training strategy, our structure can be directly incorporated into various types of generative models, such as GAN, or DDPM. In this paper, we primarily introduce the generative model based on the VAE. The process is illustrated in Figure. 2. Specifically, given the human motion sequence $X_1$, $X_2$ and $Y$, the encoder $q_\theta(Z|Y, X_1, X_2)$ is trained to encode the human motions into latent space $Z$, which maps the human motion distribution into the latent code distribution. Then the generator $G(\hat{Y}|Z, X_1, X_2)$ transforms the latent representation to the data manifold. In terms of the structure, both the encoder and generator are implemented by the GRU Chung et al. (2014) and Transformer Vaswani et al. (2017) network. The Transformer focuses on the inter-dependencies among human joints within the same time step. For the Transformer process, when modeling the human motion sequences at time $t$ ($t \in \{1, 2, \ldots, T\}$), we project the whole sequence of joint embedding $\boldsymbol{E}_t$ into matrices $\boldsymbol{Q}$, $\boldsymbol{K}$, and $\boldsymbol{V}$ by $\boldsymbol{W}_Q, \boldsymbol{W}_K, \boldsymbol{W}_V$. $\boldsymbol{Q} = \boldsymbol{E}_t \boldsymbol{W}_Q$, $\boldsymbol{K} = \boldsymbol{E}_t \boldsymbol{W}_K$, and $\boldsymbol{V} = \boldsymbol{E}_t \boldsymbol{W}_V$. The number of used human joint is $a$ ($a \in \{1, 2, \ldots, A\}$). The summary of the spatial joints $\tilde{\boldsymbol{E}}_t$ is calculated by aggregating all the joint information using the multi-head mechanism. The $GRU_\phi$ with parameter $\phi$ intends to capture the smoothness property of human motions, and then encode the human motions into latent space $Z$. The formula for using the encoder to map the human motions into latent space is computed as follows:

$$
\begin{aligned}
Attention(Q, K, V) &= softmax(\frac{QK^T}{\sqrt{d_k}})V, \\
head_i &= Attention(\boldsymbol{Q}^{(i)}, \boldsymbol{K}^{(i)}, \boldsymbol{V}^{(i)}), \\
\tilde{\boldsymbol{E}}_t &= Concat(head_1, \ldots, head_H)\boldsymbol{W}^{(O)}, \\
Z &\leftarrow GRU_\phi(\tilde{\boldsymbol{E}}_t),
\end{aligned}
\tag{7}
$$

where $W^{(O)}$ denotes the concatenation weight matrix, the attention is computed by dot products of the query $\boldsymbol{Q}$ with all keys $\boldsymbol{K}$, divide each by $\sqrt{d_k}$, and apply a softmax function to obtain the weights on the values $\boldsymbol{V}$. In addition, the architecture of the generator is the same as the encoder. The generator aims to map the latent space $Z$ back to the reconstructed human motion sequence $\hat{Y}$. To train the VAE network, the reconstruction and KL divergence loss functions are utilized to measure the difference between the reconstructed and the original motion sequences, which are

defined as follows:

$$L_{VAE} = E_{q(Z|Y,X_1,X_2)}(log(p(Y|Z,X_1,X_2)) - KL(q(Z|Y,X_1,X_2)||p(Z|Y)), \quad (8)$$

where the first term is expressed as reconstruction loss function. The second indicates the KL divergence, $q(Z|Y,X_1,X_2)$ and $p(Z|Y)$ denote the posterior and prior distribution (commonly a Gaussian distribution), respectively.

**Remark**: Although the backbone of the generator is primarily discussed in the context of a VAE, the proposed structure can be directly applied to other generative models. This can be achieved by aligning the loss functions and training details of models like GAN and DDPM with those of their standard implementations Goodfellow et al. (2020); Ho et al. (2020).

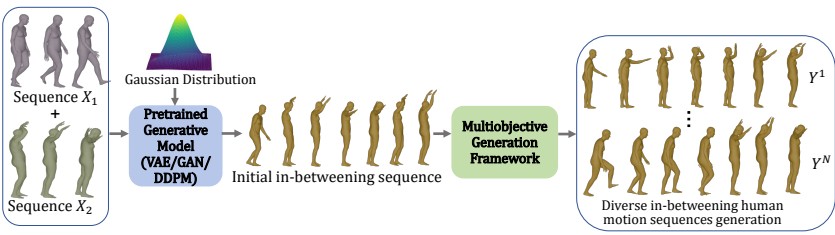

Figure 3: The framework of our method. We first randomly sample from the Gaussian distribution, and use the pretrained generator to generate the initial in-betweening motion sequence. Then, the designed multiobjective generation framework is utilized to explore multiple optimal solutions for more diverse sequences.

### 4.3 A Summary for the Proposed Method

In summary, our task aims to generate diverse and smooth in-betweening human motions given the user-provided human motion frames. The overflow of the proposed method is briefly described in Figure. 3. Specifically, given the user-provided motion sequence $X_1$ and $X_2$, we first generate the initial in-betweening human motions $Y$ through a designed and pretrained VAE model. We would like to note that other generative models can also be used for this task, such as GAN and DDPM, as our method can enhance the performance of these generative models. To alleviate the mode collapse in VAE-based generation, while enhancing the diversity and smoothness of the generated motion sequence, we transform this task into multiobjective optimization problem, and design a multiobjective optimization function to optimize this process. We have demonstrated that for any Pareto optimal solutions of the formulated problem, the corresponding human motion sequences can support a smooth and natural transition to interpolate user-provided sequences. Therefore, the multiobjective optimization process can capture multiple optimal solutions, i.e., generate diverse in-betweening human motion sequences. Moreover, our method utilizes the generative model to generate motion sequences. It is significant to note that the proposed method can enhance the diversity of generated human motions based on generative models without introducing additional training processes and parameters.

## 5 Experiments and Design

### 5.1 Datasets

The experiments are performed on four widely employed motion datasets: BABEL Punnakkal et al. (2021), HumanAct12 Guo et al. (2020), NTU RGB-D Liu et al. (2019), GRAB Taheri et al. (2020). The details can be found in Appendix.

### 5.2 Parameter Settings

For the experiment settings, the batch size for training the VAE model is set to 128 and the number of used human joints is 16. The Transformer uses 8 attention heads. The proposed method is implemented using the PyTorch framework in Python 3.6. To ensure convergence, the Adam optimizer is used to train the model, with an initial learning rate of $10^{-2}$ that decays by 0.98 every 10 epochs. The training is conducted for 500 epochs. For the multiobjective generation framework, the $I_{max}$ is

Table 1: Quantitative comparison results. $FID_{tr}$ and $FID_{te}$ refer to the FID scores obtained from the generation to train and test datasets, respectively. The best results are in bold.

| Method | BABEL | | | | | HAct12 | | | | |
|---|---|---|---|---|---|---|---|---|---|---|
| | $FID_{tr}\downarrow$ | $FID_{te}\downarrow$ | $ACC\uparrow$ | $ADE\downarrow$ | $APD\uparrow$ | $FID_{tr}\downarrow$ | $FID_{te}\downarrow$ | $ACC\uparrow$ | $ADE\downarrow$ | $APD\uparrow$ |
| RMI Harvey et al. (2020) | 37.09 | 30.15 | 1.51 | 1.21 | 0.79 | 245.35 | 298.06 | 24.51 | 1.38 | 0.60 |
| MITT Qin et al. (2022) | 32.21 | 27.10 | 0.73 | 0.99 | 0.91 | 254.72 | 143.71 | 22.73 | 1.39 | 0.53 |
| Motion DNA Zhou et al. (2020) | 27.04 | 23.25 | 16.2 | 1.12 | 0.67 | 247.68 | 139.89 | 24.46 | 1.34 | 0.87 |
| ACTOR Petrovich et al. (2021) | 29.34 | 30.31 | 40.9 | 2.29 | 2.71 | 248.81 | 381.56 | 44.41 | 1.54 | 0.95 |
| WAT (RNN) Mao et al. (2022) | 22.54 | 22.39 | 49.6 | 1.47 | 1.74 | 129.95 | 164.38 | 59.02 | 1.23 | 0.96 |
| WAT (Trans.) Mao et al. (2022) | 20.02 | 19.41 | 39.5 | 1.40 | 1.82 | 141.85 | 139.82 | 56.87 | 1.26 | 0.88 |
| MultiAct Lee et al. (2023) | 16.39 | 19.12 | 73.67 | 1.77 | 5.42 | 174.76 | 243.82 | 68.62 | 1.35 | 1.76 |
| MoFusion Dabral et al. (2023) | 15.49 | 15.12 | 74.71 | 1.07 | 6.45 | 125.41 | 134.14 | 67.14 | 1.07 | 1.45 |
| MGF-IMM (VAE) | **14.13** | 15.71 | 74.21 | 1.12 | 6.14 | **115.36** | 135.26 | 67.27 | 1.21 | 1.81 |
| MGF-IMM (GAN) | 14.46 | 14.49 | 75.16 | 1.10 | 6.96 | 116.36 | 130.49 | 68.91 | 1.20 | 1.92 |
| MGF-IMM (DDPM) | 14.32 | **13.65** | **77.32** | **1.01** | **8.45** | 116.16 | **121.34** | **70.26** | **1.17** | **2.45** |
| Method | NTU | | | | | GRAB | | | | |
| | $FID_{tr}\downarrow$ | $FID_{te}\downarrow$ | $ACC\uparrow$ | $ADE\downarrow$ | $APD\uparrow$ | $FID_{tr}\downarrow$ | $FID_{te}\downarrow$ | $ACC\uparrow$ | $ADE\downarrow$ | $APD\uparrow$ |
| RMI Harvey et al. (2020) | 144.98 | 113.61 | 66.3 | 1.11 | 1.19 | 132.28 | 102.54 | 75.1 | 1.24 | 1.79 |
| MITT Qin et al. (2022) | 151.11 | 157.54 | 70.6 | 1.20 | 1.21 | 162.34 | 148.32 | 62.1 | 1.21 | 1.19 |
| Motion DNA Zhou et al. (2020) | 147.64 | 147.92 | 68.7 | 1.19 | 1.09 | 157.99 | 146.18 | 60.7 | 1.08 | 0.97 |
| ACTOR Petrovich et al. (2021) | 355.69 | 193.58 | 66.3 | 1.49 | 2.07 | 342.73 | 181.45 | 78.1 | 1.45 | 2.13 |
| WAT (RNN) Mao et al. (2022) | 72.18 | 111.01 | 76.0 | 1.20 | 2.20 | 70.12 | 101.24 | 76.0 | 1.18 | 2.05 |
| WAT (Trans.) Mao et al. (2022) | 83.14 | 114.62 | 71.3 | 1.23 | 2.19 | 79.17 | 114.52 | 69.1 | 1.23 | 2.19 |
| MultiAct Lee et al. (2023) | 374.73 | 530.09 | 63.9 | 1.41 | 2.73 | 374.73 | 530.09 | 64.3 | 1.32 | 2.54 |
| MoFusion Dabral et al. (2023) | 75.46 | 108.12 | 77.71 | 1.06 | 2.75 | 68.12 | 95.01 | 79.21 | 1.09 | 2.01 |
| MGF-IMM (VAE) | **81.64** | **107.16** | 74.9 | 1.09 | 2.72 | 69.46 | 98.16 | 78.5 | 1.13 | 2.37 |
| MGF-IMM (GAN) | 79.47 | 107.51 | 76.4 | 1.05 | 2.86 | 67.19 | 96.49 | 80.1 | 1.08 | 2.42 |
| MGF-IMM (DDPM) | **71.90** | 107.32 | **79.8** | **1.03** | **2.94** | **65.32** | **94.72** | **85.9** | **1.04** | **2.59** |

Table 2: Ablation studies for the influence of different lengths on the BABEL dataset.

| | Fixed length $N = 20$ | | | | | Variable length | | | | |
|---|---|---|---|---|---|---|---|---|---|---|
| | $FID_{tr}\downarrow$ | $FID_{te}\downarrow$ | $ACC\uparrow$ | $ADE\downarrow$ | $APD\uparrow$ | $FID_{tr}\downarrow$ | $FID_{te}\downarrow$ | $ACC\uparrow$ | $ADE\downarrow$ | $APD\uparrow$ |
| RMI Harvey et al. (2020) | 37.09 | 30.15 | 1.51 | 1.21 | 0.79 | 36.87 | 30.03 | 2.91 | 0.59 | 1.51 |
| MITT Qin et al. (2022) | 32.21 | 27.10 | 0.73 | 0.99 | 0.91 | 31.62 | 26.64 | 1.66 | 0.65 | 1.61 |
| CMIB Kim et al. (2022) | 21.07 | 23.60 | 1.96 | 1.21 | 2.46 | 20.42 | 21.34 | 2.04 | 0.91 | 2.94 |
| MultiAct Lee et al. (2023) | 16.39 | 19.12 | 73.70 | 1.77 | 3.01 | 15.98 | 13.62 | 73.99 | 0.61 | 5.42 |
| MGF-IMM (DDPM) | 14.91 | 13.72 | 75.81 | 1.09 | 7.24 | 14.32 | 13.65 | 77.32 | 1.01 | 8.45 |

set to 20, the nondominated sorting and the crowding distance Deb et al. (2002) are utilized in the environment selection. The population size is set to 20. All the inference processes are conducted on the NVIDIA Tesla A100 GPU. The keyframes of the user-provided sequence are set to 5 in our experiment, and can be variable according to different tasks.

## 5.3 METRICS

The accuracy and diversity of the generated motion sequences are essential for in-betweening motion task. For comparison, we employ the metrics to facilitate the evaluation of MGF-IMM, i.e., *Frechet Inception Distance* (FID), *Action Accuracy* (ACC) and *Average Displacement Error* (ADE). The details about these metrics can be found in Appendix. Specifically, APD aims to evaluate the diversity performance of MGF-IMM. ACC, FID and ADE aim to evaluate the accuracy performance of MGF-IMM. For APD and ACC metrics, a higher value is better. For FID and ADE metrics, a lower value is better.

## 5.4 BASELINE METHODS

In this work, we compare the proposed method with the state-of-the-art human motion in-betweening generation methods as baseline methods, including RMI Harvey et al. (2020), MITT Qin et al. (2022), Motion DNA Zhou et al. (2020), ACTOR Petrovich et al. (2021), WAT Mao et al. (2022), MultiAct Lee et al. (2023), MoFusion Dabral et al. (2023).

## 6 RESULTS AND ANALYSIS

### 6.1 QUANTITATIVE RESULTS

Table 1 summarizes the comparative performance of the proposed MGF-IMM method against various baselines for in-betweening motion generation task. The results demonstrate that the MGF-IMM method achieves state-of-the-art performance across all evaluation metrics. Specifically, the diversity metric (APD) improvements are more obvious than the accuracy metrics. This superiority highlights the effectiveness of the multiobjective optimization framework. By formulating the in-betweening task as a multiobjective problem, MGF-IMM can generate motion sequences that

Table 3: Ablation studies for the influence of the multiobjective generation framework on the performance of our methods. *w.o.* means "without", *w.* means "with".

| Method | | BABEL | | | | | HAct12 | | | | |
|---|---|---|---|---|---|---|---|---|---|---|---|
| | | $FID_{tr}\downarrow$ | $FID_{te}\downarrow$ | $ACC\uparrow$ | $ADE\downarrow$ | $APD\uparrow$ | $FID_{tr}\downarrow$ | $FID_{te}\downarrow$ | $ACC\uparrow$ | $ADE\downarrow$ | $APD\uparrow$ |
| MGF-IMM (VAE) | *w.o.* EMG | 16.45 | 18.45 | 73.14 | 1.42 | 3.46 | 134.27 | 147.62 | 58.14 | 1.36 | 0.86 |
| | *w.* EMG | 14.13 | 15.71 | 74.21 | 1.12 | 6.14 | 115.36 | 135.26 | 67.27 | 1.21 | 1.81 |
| MGF-IMM (GAN) | *w.o.* EMG | 16.13 | 17.16 | 73.56 | 1.36 | 4.02 | 125.86 | 137.94 | 66.13 | 1.35 | 1.74 |
| | *w.* EMG | 14.46 | 14.49 | 75.16 | 1.10 | 6.96 | 116.36 | 130.49 | 68.91 | 1.20 | 1.92 |
| MGF-IMM (DDPM) | *w.o.* EMG | 15.51 | 14.61 | 75.04 | 1.07 | 6.84 | 124.65 | 131.74 | 67.46 | 1.36 | 1.84 |
| | *w.* EMG | 14.32 | 13.65 | 77.32 | 1.04 | 8.45 | 116.16 | 121.34 | 70.26 | 1.17 | 2.45 |

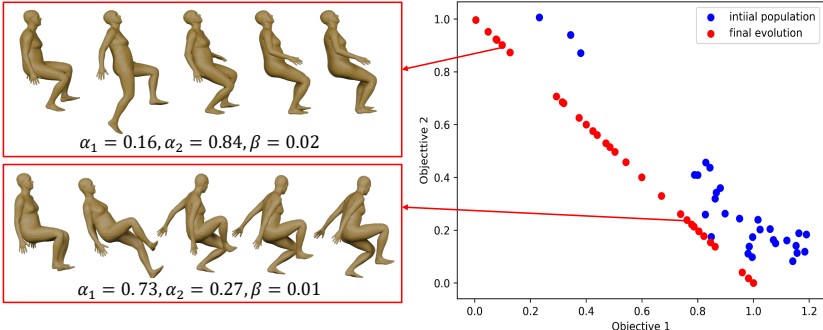

$\alpha_1 = 0.16, \alpha_2 = 0.84, \beta = 0.02$

$\alpha_1 = 0.73, \alpha_2 = 0.27, \beta = 0.01$

Figure 4: Pareto front of the optimization process under the "Sit" motion sequences. The blue points denote the initial population. The red points denote the final population.

satisfy the proposed different objectives. Interestingly, the MGF-IMM method exhibits relatively weaker performance on the NTU RGB-D dataset compared to other datasets. This is likely due to the higher level of noise and artifacts present in NTU RGB-D, which poses additional challenges for MGF-IMM. On the other datasets, the MGF-IMM method demonstrates substantial improvements in both the accuracy and diversity metrics. This is a direct result of the multiobjective optimization approach, which can explore and provide a diverse set of solutions between the competing objective functions. Overall, the comprehensive evaluation on multiple human motion datasets shows the superiority of the proposed MGF-IMM method for in-betweening human motion generation task. The multiobjective optimization function proves to be a highly effective technique for this task.

In addition, the results in Table 1 also report the performance of MGF-IMM using different generative models. Specifically, the table compares MGF-IMM when using VAE, GAN, and DDPM as the base generative models. The results reveal that the MGF-IMM approach consistently outperforms other in-betweening methods regardless of the specific generative model. This indicates that the multiobjective optimization method introduced in MGF-IMM effectively enhances performance of the generated human motions.

Table 4: Ablation studies for the influence of the intra-class difference. *w.o.* means "without", *w.* means "with".

| Method | | BABEL | | | | | HAct12 | | | | |
|---|---|---|---|---|---|---|---|---|---|---|---|
| | | $FID_{tr}\downarrow$ | $FID_{te}\downarrow$ | $ACC\uparrow$ | $ADE\downarrow$ | $APD\uparrow$ | $FID_{tr}\downarrow$ | $FID_{te}\downarrow$ | $ACC\uparrow$ | $ADE\downarrow$ | $APD\uparrow$ |
| MGF-IMM (VAE) | *w.o.* intra-class | 15.94 | 17.04 | 73.46 | 1.34 | 5.97 | 129.27 | 142.64 | 61.14 | 1.35 | 1.58 |
| | *w.* intra-class | 14.13 | 15.71 | 74.21 | 1.12 | 6.14 | 115.36 | 135.26 | 67.27 | 1.21 | 1.81 |
| MGF-IMM (GAN) | *w.o.* intra-class | 16.24 | 16.42 | 73.95 | 1.29 | 6.46 | 124.36 | 133.17 | 66.13 | 1.29 | 1.76 |
| | *w.* intra-class | 14.46 | 14.49 | 75.16 | 1.10 | 6.96 | 116.16 | 130.49 | 68.91 | 1.20 | 1.92 |
| MGF-IMM (DDPM) | *w.o.* intra-class | 15.03 | 14.19 | 76.36 | 1.06 | 7.49 | 121.45 | 129.58 | 67.42 | 1.64 | 2.14 |
| | *w.* intra-class | 14.32 | 13.65 | 77.32 | 1.04 | 8.45 | 116.16 | 121.34 | 70.26 | 1.17 | 2.45 |

## 6.2 ABLATION STUDIES

The paper conducts ablation studies to analyze the contributions of the different components within the proposed MGF-IMM method. As shown in Table 2, experiments compare the influence of different methods under variable and fixed length of transition sequences, including RMI Harvey et al. (2020), MITT Qin et al. (2022), MultiAct Lee et al. (2023) and CMIB Kim et al. (2022). The comparison results clearly show that allowing variable length transition sequence generation leads to enhanced performance compared to the fixed length. This indicates the importance of flexibility in the length of in-betweening motions for achieving high-quality results.

Table 3 aims to assess the impact of the introduced multiobjective generation framework. Specifically, we evaluated the performance of the multiobjective generation framework combined with various generative models, including VAE, GAN, and DDPM. The results show that the inclusion of

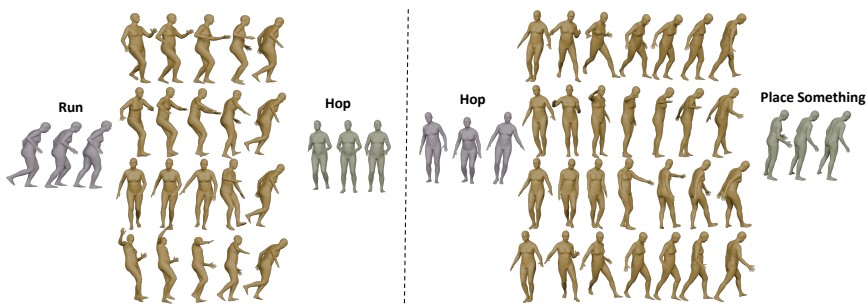

Figure 5: The generated diverse in-betweening human motion sequences.

the multiobjective generation framework consistently improves performance, particularly in terms of the diversity of the generated motion sequences within a bath process. These ablation findings further report the effectiveness of our design choices.

In addition, Table 4 reports the influence of the intra-class difference within the same human motions based on different generative models. The second term of $\alpha_1$ in the objective function 2 aims to capture the intra-class difference within the same motion, which denotes different probabilities corresponding to the same action label. From the comparison results, we can observe that introducing this intra-class difference term into the optimization process can effectively enhance the performance of the generated motion sequences. This finding highlights the importance of explicitly modeling the inherent variations within the same motion class.

### 6.3 QUALITATIVE RESULTS

In the qualitative analysis, we first visualize the Pareto front obtained from the multiobjective optimization process. Figure. 4 plots the distribution of the 20 population solutions. The blue points denote the initial population, which is the generated motion sequence without the multiobjective generation framework. The red points represent the generated in-betweening human motion sequences after the optimization process. As shown in this figure, the initial population is unable to approximate the Pareto front effectively, and the distribution of the solutions is rather irregular. In contrast, the MGF-IMM approach is much more effective in locating multiple optimal solutions along the Pareto front. In addition, we show two points on the Pareto front that are far apart, which correspond to two human motion sequences. This demonstrates its superior ability to balance the trade-offs between the competing objectives during the in-betweening human generation task.

In addition, we also visualize the in-betweening human motion sequences given the user-provided motion sequences. As illustrated in Figure. 5, the left side of the figure shows the transition from the "Run" to the "Hop" action, while the right side displays the transition from the "Hop" to the "Place" action. Since the two actions on the left have a small difference, the generated transition motion sequence is relatively short. The comparison results show that our method can generate variable lengths in-betweening human motion sequences. These qualitative results demonstrate that our method can generate more diverse human motion sequences under the introduced multiobjective generation framework.

## 7 CONCLUSION

This paper introduces the multiobjective optimization into the generative models and takes the in-betweening human motion task as a showcase to report the effectiveness of the proposed method MGF-IMM. We transform in-betweening human motion task into a multiobjective optimization problem, and design a multiobjective function to enhance the diversity and smoothness of the generated motion sequences. In addition, multiobjective EA is utilized to explore the Pareto front solutions. The designed generative model is utilized to generate the motion sequence. Note that the multiobjective generation framework works in the inference process, therefore, the proposed method is capable of enhancing the diversity and accuracy of this task without introducing additional parameters. Comprehensive experiments have been conducted to demonstrate the effectiveness of the proposed method.

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

APPENDIX

## A  PROOF OF PROPOSITION 1

The proof of *Theorem 1*.

Assuming that there exist an in-betweening motion sequence $Y_b^*$ that can Pareto dominate $Y_b$ in problem equation 4, then we have:

$$F_1(Y_b^*) < F_1(Y_b), F_2(Y_b^*) < F_2(Y_b). \tag{9}$$

Thus, we can obtain that

$$
\begin{aligned}
&F_1(Y_b^*) + F_2(Y_b^*) < F_1(Y_b) + F_2(Y_b) \\
\Rightarrow &\alpha_1(Y_b^*) + \alpha_2(Y_b^*) + 2\beta(Y_b^*) < \alpha_1(Y_b) + \alpha_2(Y_b) + 2\beta(Y_b) \\
\Rightarrow &1 + 2\beta(Y_b^*) < 1 + 2\beta(Y_b) \\
\Rightarrow &\beta(Y_b^*) < \beta(Y_b),
\end{aligned}
\tag{10}
$$

where line 2 and line 3 in equation 10 are obtained based on the equation 2. It should be noticed that, we have $\beta(Y_b^*) < \beta(Y_b)$ from equation 10, and this contradicts the assumption that $Y_b \in \mathcal{B}$. Therefore, there is no in-betweening human motion sequence that can Pareto dominate $Y_b$, which demonstrates our proof.

## B  PROOF OF PROPOSITION 2

The proof of *Theorem 2*.

Considering that $Y_1, Y_2 \in \mathcal{B}$, we have $\beta(Y_1) = \beta(Y_2) = \min_Y \beta(Y)$. According to equation 4, we have the following equations:

$$
\begin{aligned}
F_1(Y_1) - F_1(Y_2) &= \frac{1}{D}\left(C(Y_1) + P_c(Y_1) - C(Y_2) - P_c(Y_2)\right) \\
F_2(Y_1) - F_2(Y_2) &= -\frac{1}{D}(C(Y_1) + P_c(Y_1)) + \frac{1}{D}(C(Y_2) + P_c(Y_2))
\end{aligned}
\tag{11}
$$

Then, we can obtain that

$$
\begin{aligned}
&||\mathbf{F}(Y_1) - \mathbf{F}(Y_2)|| \\
=&|F_1(Y_1) - F_1(Y_2)| + |F_2(Y_1) - F_2(Y_2)| > \frac{4}{D} \\
\Rightarrow &\left|\frac{1}{D}\left(C(Y_1) + P_c(Y_1) - C(Y_2) - P_c(Y_2)\right)\right| + \left|\frac{1}{D}(C(Y_2) + P_c(Y_2) - C(Y_1) - P_c(Y_1))\right| > \frac{4}{D} \\
\Rightarrow &2\left|\frac{1}{D}\left(C(Y_1) + P_c(Y_1) - C(Y_2) - P_c(Y_2)\right)\right| > \frac{4}{D} \\
\Rightarrow &|C(Y_1) + P_c(Y_1) - C(Y_2) - P_c(Y_2)| > 2 \\
\Rightarrow &|C(Y_1) - C(Y_2)| > 2 - |P_c(Y_1) - P_c(Y_2)|
\end{aligned}
\tag{12}
$$

Considering that both $P_c(Y_1)$ and $P_c(Y_2)$ are probabilities, i.e., $P_c(Y_1) \in [0, 1]$ and $P_c(Y_2) \in [0, 1]$, thus we have

$$|P_c(Y_1) - P_c(Y_2)| \le 1, \tag{13}$$

thus resulting in $|C(Y_1) - C(Y_2)| > 1$. Given that $C(Y)$ is the result of the classifier $C$ applied to sequence $Y$ and that its value is an integer, we can obtain the conclusion that $|C(Y_1) - C(Y_2)| > 1$ is equivalent to $C(Y_1) \ne C(Y_2)$. This shows that sequence $Y_1$ and sequence $Y_2$ correspond to different kinds of motions.

## C  DATASET

**BABEL** is a subset of the AMASS dataset with per-frame action annotations. In this work, we split the dataset into two parts: single-action sequences and transition sequences between two actions. We downsample all motion sequences to 30 Hz. For single-action motions, we divide the long motions into several short ones. Each short motion performs one single action, and the remove too short sequences ($< 1$ second). We also eliminate the action labels with too few samples ($< 60$) or overlap with other actions. After processing, we have 20 action categories.

**HumanAct12** contains 12 subjects in which 12 categories of actions with per-sequence annotation are provided. The sequences with less than 35 frames are removed, which results in 727 training and 197 testing sequences. Subjects P1 to P10 are used for training, P11 and P12 are used for testing.

**NTU RGB-D** originally contains over 100,000 motions with 120 classes whose pose annotations are from MS Kinect readout, which makes the data highly noisy and inaccurate. To facilitate training, our operation is consistent with the operation in Kocabas et al. (2020), which re-estimates the 3D positions of 18 body joints (i.e. 19 bones) from the point cloud formed by aligning synchronized video feeds from multiple cameras. Note the poses are not necessarily matched perfectly to their true poses. It is sufficient here to be perceptually natural and realistic.

**GRAB** consists of 10 subjects interacting with 51 different objects, performing 29 different actions. Since, for most actions, the number of samples is too small for training, we choose the four action categories with the most motion samples, i.e., Pass, Lift, Inspect and Drink. We use 8 subjects (S1-S6, S9, S10) for training and the remaining 2 subjects (S7, S8) for testing. In all cases, we remove the global translation. The original frame rate is 120 Hz. To further enlarge the size of the dataset, we downsample the sequences to 15-30 Hz.

## D  METRIC DEFINITIONS

1. *Frechet Inception Distance* (FID). FID is the distribution similarity between the predicted sequences and the ground-truth motions:

$$FID = \|\mu_{\text{gen}} - \mu_{\text{gt}}\|^2 + Tr(\Sigma_{\text{gen}} + \Sigma_{\text{gt}} - 2(\Sigma_{\text{gen}}\Sigma_{\text{gt}})^{1/2}), \qquad (14)$$

where $\mu \in \mathbb{R}^F$ and $\sigma \in \mathbb{R}^{F \times F}$ are the mean and covariance matrix of perception features obtained from a pretrained motion classifier model with $F$ dimension of the perception features. $Tr(\cdot)$ denotes the trace of a matrix. In the experiment, we report the FID of generation to train ($FID_{tr}$) and test ($FID_{te}$) datasets, respectively.

2. *Action Accuracy* (ACC). To evaluate motion realism, we report the action recognition accuracy of the generated motions using the same pretrained action recognition model.

3. *Average Displacement Error* (ADE). ADE is the $L2$ distance between the predicted motion and ground-truth motion to measure the accuracy of the whole sequence:

$$ADE = \min_{i=(1,2,3,\cdots,N)} \frac{1}{T} \sum_{k=1}^{T} \|\hat{\mathbf{Y}}_k^i - \mathbf{Y}_k^i\|_2, \qquad (15)$$

where $Y$ is the Ground Truth motion sequence. $\hat{Y}$ is the predicted motion sequence. $N$ is the final number of predicted motion sequences.

4. *Average Pairwise Distance* (APD). APD is the average $L2$ distance between all the generation pairs, it is used to measure the diversity:

$$APD = \frac{1}{N(N-1)} \sum_{i=1}^{N} \sum_{j=1,j\neq i}^{N} \|(\hat{\mathbf{Y}})^i - (\hat{\mathbf{Y}})^j\|, \qquad (16)$$

## E  ADDITIONAL EXPLANATION OF THE INTRODUCED MULTIOBJECTIVE MODELING

Although we provide two theorems in Section 3 to elucidate the capability of our multi-objective modeling method to preserve the diversity of a sequence set, these theorems may not be immedi-

ately intuitive. For readers without a background in multi-objective optimization, this section might pose significant challenges. To improve comprehension, we present two examples that illustrate the meaning of the Pareto dominance relationship and the principle of diversity preservation in the proposed method. Before proceeding, it is important to clarify that our approach diverges from the conventional idea which uses a *single indicator function* to evaluate and enhance diversity within a batch of human motion sequences. Instead, we translate the diversity of the sequence batch into diversity in the objective space of multi-objective optimization. Leveraging the well-established ability of evolutionary algorithms to maintain diversity in the objective space, our method effectively generates batches of diverse motion sequences.

### E.1   PARETO DOMINANCE RELATIONSHIP

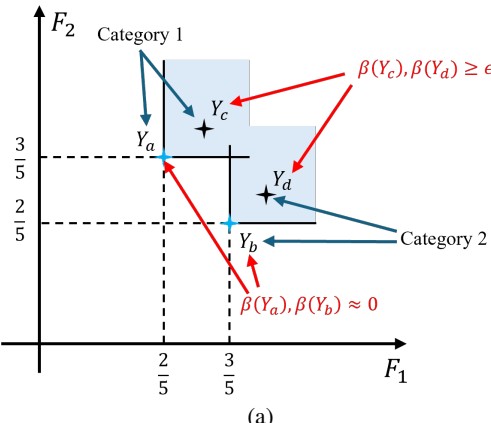

(a)

Figure 6: Examples of Pareto dominance relationship in the proposed multi-objective modeling approach. In the example, we assume there are motion sequences $Y_a$, $Y_b$, $Y_c$, and $Y_d$. For $Y_a$ and $Y_b$, we assume that $C(Y_a) = 1$, $C(Y_b) = 2$, $P_c(Y') = 1$, and $\beta(Y') = 0$, $Y' \in \{Y_a, Y_b\}$). Meanwhile, for $Y_c$ and $Y_d$, we assume that $C(Y_c) = 1$, $C(Y_d) = 2$, $P_c(Y') = 1$, and $\beta(Y') \geq \epsilon$, $Y' \in \{Y_c, Y_d\}$), where $\epsilon$ is a positive value. The objective space illustrates that the full smooth motion sequences Pareto dominates non-full smooth sequence, i.e., $Y_a$ Pareto dominates $Y_c$, and $Y_b$ Pareto dominates $Y_d$.

First, we introduce the concept of Pareto dominance in the context of the proposed multi-objective optimization problem. Assume there are four motion sequences, $Y_a$, $Y_b$, $Y_c$, and $Y_d$. Specifically, $Y_a$ and $Y_b$ are motion sequences belonging to the first and second categories, respectively (i.e., $C(Y_a) = 1$ and $C(Y_b) = 2$), with absolute category memberships (i.e., $P_c(Y') = 1$, $Y' \in \{Y_a, Y_b\}$). Furthermore, $Y_a$ and $Y_b$ are assumed to exhibit full smoothness (i.e., $\beta(Y') = 0$, $Y' \in \{Y_a, Y_b\}$). Additionally, consider two other motion sequences, $Y_c$ and $Y_d$, which satisfy $C(Y_c) = 1$ and $C(Y_d) = 2$, but cannot guarantee full smoothness (i.e., $\beta(Y') \geq \epsilon$, $Y' \in \{Y_c, Y_d\}$), where $\epsilon$ is a positive value). As shown in Figure 6, the objective space illustrates that $Y_a$ Pareto dominates $Y_c$ and $Y_b$ Pareto dominates $Y_d$. This is because the objective function values corresponding to $Y_a$ and $Y_b$ are superior to those of $Y_c$ and $Y_d$, respectively. Evidently, the Pareto dominance outcome is influenced by whether the motion sequence ensures complete smoothness. The smaller the $\beta$ function value, which determines smoothness, the less likely the motion sequence is to be dominated in the objective space. Therefore, for the multi-objective optimization problem we have defined, obtaining Pareto-optimal sequences inherently favors the generation of smooth motion sequences.

### E.2   DIVERSITY KEEPING FROM THE OBJECTIVE SPACE

We now provide an example to demonstrate how diversity in the objective space translates into the intra-batch diversity. Consider two batches, $\mathcal{Y}_1 = \{Y_1^{(1)}, \dots, Y_1^{(5)}\}$ and $\mathcal{Y}_2 = \{Y_2^{(1)}, \dots, Y_2^{(5)}\}$, each consisting of five motion sequences. In $\mathcal{Y}_1$, the contained sequences belong to five different motion categories, whereas in $\mathcal{Y}_2$, all sequences belong to a single category (category 0). Clearly, $\mathcal{Y}_1$ exhibits significantly greater diversity than $\mathcal{Y}_2$. Assume that for all sequences in $\mathcal{Y}_1$ and $\mathcal{Y}_2$, their category membership is absolute ($P_c(Y') = 1$, $Y' \in \mathcal{Y}_1 \cup \mathcal{Y}_2$), and smoothness is fully guaranteed ($\beta(Y') = 0$, $Y' \in \mathcal{Y}_1 \cup \mathcal{Y}_2$). The representations of these two batches in the objective space are

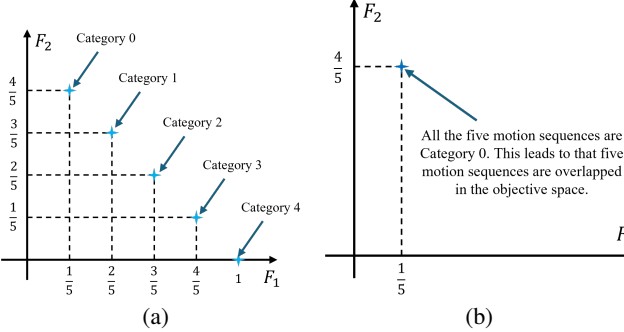

(a)                                                 (b)

Figure 7: Examples of diversity preservation in the proposed multi-objective modeling approach. In the example, we show the representation of motion sequences in the objective spaces corresponding to the constructed multi-objective optimization problem. We assuming there are two batches, $\mathcal{Y}_1 = \{Y_1^{(1)}, \ldots, Y_1^{(5)}\}$ and $\mathcal{Y}_2 = \{Y_2^{(1)}, \ldots, Y_2^{(5)}\}$, each consisting of five motion sequences. In $\mathcal{Y}_1$, the contained sequences belong to five different motion categories, whereas in $\mathcal{Y}_2$, all sequences belong to a single category (category 0). Assume that for all sequences in $\mathcal{Y}_1$ and $\mathcal{Y}_2$, their category membership is absolute, i.e., $P_c(Y') = 1, Y' \in \mathcal{Y}_1 \cup \mathcal{Y}_2$, and smoothness is fully guaranteed, i.e., $\beta(Y') = 0, Y' \in \mathcal{Y}_1 \cup \mathcal{Y}_2$. (a) Representation of motion sequences in the objective space for the batch $\mathcal{Y}_1$. (b) Representation of motion sequences in the objective space for the batch $\mathcal{Y}_2$.

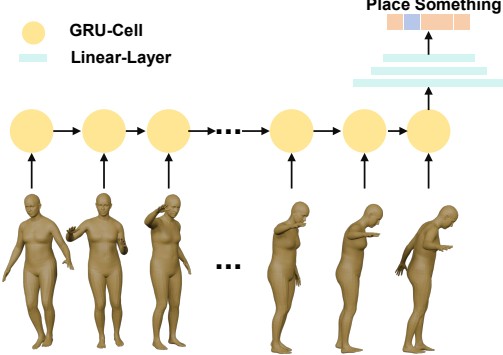

Figure 8: The illustration of multi-class classifier model.

depicted in Figure 7. As shown in Figure 7(a), the batch $\mathcal{Y}_1$, characterized by high diversity, also demonstrates excellent diversity in the objective space, comprehensively covering the Pareto front of the established multi-objective optimization problem. In contrast, for $\mathcal{Y}_2$, all five sequences correspond to overlapping points in the objective space, as illustrated in Figure 7(b). This indicates that for batches with poor diversity, like $\mathcal{Y}_2$, maintaining diversity in the objective space fails. The above example illustrates that for the multi-objective optimization problem we have formulated, diversity preservation in the objective space translates into motion sequence diversity within the batch itself. Maintaining diversity in the objective space has been extensively studied in the field of evolutionary algorithms, with well-established strategies. By integrating our multi-objective optimization problem with these evolutionary algorithm techniques, the proposed method effectively ensures intra-batch diversity.

## F    MULTI-CLASS CLASSIFIER MODEL

In Figure 8, we provide the network structure of the multi-class classifier model used for the *Diversity Component*. This model consists of GRU layers to encode the temporal information and MLP layers to produce the final classification results. The model is then trained for 500 epochs using the ADAM optimizer with an initial learning rate of 0.001.

