# OpenReview forum: "Novelty Unlocking with Multiobjective Generative Models: Batch Diversity of Human Motions"
_ICLR.cc/2025/Conference — Submitted to ICLR 2025_

### Official Review · Reviewer_jtPA · 2024-10-19

**Soundness:** 3
**Presentation:** 3
**Contribution:** 3
**Rating:** 6
**Confidence:** 3

**Summary:**

This paper addresses the problem of generating diverse in-between motions by formulating it as a multi-objective optimization problem. The proposed Multiobjective Generation Framework with In-Betweening Motion Model introduces an evolutionary algorithm at the inference stage to enhance motion diversity while maintaining motion quality.

**Strengths:**

The authors' formulation of the diverse motion generation problem as a multi-objective optimization problem is novel. The derived general generative framework effectively guides the sampling process towards diverse motions, successfully balancing diversity and quality in generated in-betweening human motion sequences.

**Weaknesses:**

Conditional diffusion models, such as those presented in [1] seem to solve the same problem using generative models with conditional diffusion. By changing the conditions, they can also generate diverse motions. It would be helpful if the authors could compare their approach to these methods or explain why the multiobjective formulation is necessary, and whether it can be further combined with conditional motion generators. Moreover, a discussion over the difference and advantage in terms of formulation compared with [1] would enhance the quality of the paper.

[1] S. Cohan, G. Tevet, D. Reda, X. B. Peng, and M. van de Panne, “Flexible Motion In-betweening with Diffusion Models,” in ACM SIGGRAPH 2024 Conference Papers, in SIGGRAPH ’24. New York, NY, USA: Association for Computing Machinery, Jul. 2024, pp. 1–9. doi: 10.1145/3641519.3657414.

**Questions:**

1. **Computational Overhead of the Evolutionary Algorithm:** EA will introduce extra computational cost during inference time, especially when using expensive diffusion models. Could the authors elaborate on what the computational overhead is compared with a stand-alone generative model?


2. **Comparison with Conditional Generative Models:** Given the classifier that the authors trained, one might imagine training a conditional generative model based on the classifier. Could the authors explain why using an EA would be better than training a motion-class-conditioned generative model?

---

> ### Author Response · Authors · 2024-11-20
> **Rebuttal to Reviewer jtPA**
>
> **W1, Difference from Conditional Generation-Based Methods and Universality of the Proposed Method**:
>
> We agree with your observation that, given a conditional generative model, diverse motion sequences can be obtained by providing different conditions to the model. However, it is important to emphasize that our method does not conflict with conditional generative models; rather, they can be effectively combined to achieve even greater diversity in motion sequences. A straightforward approach is to first use our multi-objective based generation approach to produce a batch of motion sequences under a fixed set of conditions. Then, by altering the conditions and reapplying our method, additional batches of motion sequences can be generated. As discussed in the overall rebuttal about the advantages, our method offers a convenient, non-training, plug-and-play framework. We believe this practicality makes our approach particularly valuable for real-world applications. Following your suggestion, we have also cited the recommended paper in the revised version.
>
> **Q1, Computational Consumption of the Evolutionary Algorithm:**
> I agree with your observation that evolutionary algorithms inherently rely on an iterative process. However, as discussed in our overall rebuttal:
> 1. Compared to other iterative approaches such as rejection sampling, our method generates more diverse and accurate sequences within the same number of iterations. This means that, for sequences of comparable quality, our approach requires significantly less sampling time than other iteration-based methods. This is substantiated by the additional experiments detailed in the overall rebuttal.
> 2. Multi-step generative processes like DDPM inherently involve multiple steps and longer inference times. However, advancements such as DDIM have been shown to significantly reduce inference durations. While DDIM may compromise generation quality and diversity, incorporating our method within DDIM mitigates these limitations, improving diversity without substantially increasing inference time compared to DDPM. The results demonstrate that when combined with DDIM, the time required by our approach remains short and comparable to that of DDPM.
>
> These findings collectively highlight the practicality and efficiency of our method for real-world applications, demonstrating its ability to balance diversity, quality, and computational cost effectively.
>
> **Q2, Comparison with Conditional Generative Models:**
> We agree that we can train a conditional motion generation method using the pretrained motion classifier, then the task is transformed into action-conditioned human motion generation. In this way, an additional *action-conditional motion generation model* has to be trained for this task, which introduces an additional training process. It is noted that the action-conditioned generation is different from our task. If we utilize the action label to generate the in-betweening motion sequences, then the generated sequence belongs to a specific motion pattern, in contrast, our method encourages the diversity of the generated motion patterns. Therefore, the pretrained classifier is more appropriate as a tool to enhance diversity.
>
> In addition, the designed EA process aims to optimize the multiobjective process, which can enhance the diversity and quality through the designed multiobjective function. This operation can improve the existing generative model without introducing additional training parameters, as we explain in the response to **W1**.

---

> ### Author Response · Authors · 2024-11-24
> **Look forward to hearing your thoughts**
>
> Dear Reviewer,
> We hope the explanations and responses we provided have addressed your concerns. If there are any remaining questions or areas of confusion, we would be more than happy to continue the discussion and provide further clarification. Your feedback is invaluable to us, and we look forward to hearing your thoughts.

---

> > ### Comment · Reviewer_jtPA · 2024-11-25
> > **Reply to rebuttal**
> >
> > Thanks for the author's clarification of the relationship between their multi-objective learning approach and the conditional diffusion approach. I agree that those two approaches can be complementary, and maybe that could be a meaningful future direction.

---

> > > ### Author Response · Authors · 2024-11-25
> > > **Thanks**
> > >
> > > Dear Reviewer jtPA, thanks for your positive support!

---

### Official Review · Reviewer_MnFJ · 2024-11-03

**Soundness:** 3
**Presentation:** 2
**Contribution:** 2
**Rating:** 5
**Confidence:** 4

**Summary:**

A method is presented for generating batches of diverse examples for motion in-between problems, via generative modeling,
i.e., a VAE, GAN, or DDPM model.  Thus given two motion sequences A and B, the goal is to be able to generate diverse sequences
(possibly of varying length) that connect the end of A to the start of B. This is formulated as a multiobjective optimization problem,
whereby a population of samples is repeatedly adapted to capture the pareto front of this optimization.
Key to the method is a classifier C(Y) that can categorize the motion-type of a motion Y into one of D distinct categories.
The multiobjective optimization is defined in a 2D space defined by two functions, F1(Y) = alpha1(Y) + Beta(Y)
and F2(Y) = (1-alpha1(Y)) + Beta(Y).
Alpha1(Y) maps the most-likely class evenly along the interval [0,1], (plus an epsilon that is related to the likelihood of that class).
Beta(Y) defines a smoothness loss.  Points along the pareto front thus seek to maximize class diversity, as well as being as
smooth as possible.  Diverse samples are generated by iteratively calling the generative network, each time conditioning
the network on the motions to be connected, e.g., A and B as above, and the already-generated samples, Y_i.
A memory-augmented Transformer-based architecture is used as the encoder for a conditional VAE, to generate new samples.
The results also present GAN and DDPM generative results.

The method is trained and tested on 4 different motion datasets (BABEL, HAct12, NTU, GRAB) and evaluated
based on FID and a diversity metric (APD), and two others (ACC, ADE).
The method produced the best results for FID and diversity, particularly for the DDPM version.
In addition to the quantitative results, qualitative results are provided using a visualization of the pareto front,
and a figure that illustrates 4 diverse samples generated for 2 example problems.
No video is included.

**Strengths:**

- an interesting problem, i.e., maximizing diversity within a batch of samples coming from a generative motion model.
  The specific problem being tackled here, i.e., generating diverse ways to produce a bridging motion that connects
two existing motion sequences is interesting, although it is quite a specific problem domain.
- Reasonablly extensive quantitative evaluations
- Potentially an interesting and novel method for achieving the diversity, although I still do not understand it all
- the applicability of this as a method to different classes of generative models is a real plus, i.e.,
  GANs, VAEs, and diffusion models.

**Weaknesses:**

Weaknesses:
- It is difficult to understand the given multiobjective framework.
  As best I understand it, is described by a 2D pareto optimal front, as described in the summary I've given above.
  In particular, the intuition of equation should be described. Figures and diagrams would really help the exposition.
  The specific motivation and geometric intuition for alpha1(Y) could be explained, and how it encourages diversity.
  This reader spent was stuck on equation (2) for a long amount of time.
- The multi-class classifier C(Y) is key to the method, but is not described in detail in the results and experiments,
  unless I missed it. This leaves the reader confused about the intent of this classifier, and its importance.
- There are no video results, unless I missed it as supplemental material (I did check OpenReview twice for this).
  This makes it really difficult to judge the quality and diversity of the output in practice.
- A variety of things about the method that were difficult for this reader to understand -- see questions below.

Minor comments:
- The structure of the paper was challenging for this reader.  Leading with Figure 3, then Figure 2, and then Figure 1
  is an alternate order that makes more sense to this reader.
- L047: "batch diversity" is ambiguous, i.e., it could mean intra-batch or inter-batch diversity.
- the notation of "decision variables" comes as a surprise.  Perhaps it is standard in multiobjective optimization,
  but it is worthwhile motivating in the current context.  Is it simpler to be thinking in terms of "samples"
  and "sample space"?
- the Pareto Dominance math is more intuitive when helped by a related figure
- related works on animation and pareto-frontiers:
   "Sampling of Pareto-Optimal Trajectories using Progressive Objective Evaluation in Multi-Objective Motion Planning"
   "Diverse motion variations for physics-based character animation"
- "in-betweenning" should probably be "in-betweening"
- eqn (4):  the addition of beta(Y) here seems like a bit of a hack.
  It would also be nice to more generally understand why the smoothness component is needed, i.e., why can't the
  conditional generative model capture this?  Also, Beta(Y) in eqn (3) appears to be a vector, whereas in eqn(4) it is scalar.
  Which is it?
- "the generative model is prompted":  in the age of LLMs, this is an overloaded (and therefore ambiguous) phrasing,
  as there are no LLMs involved.
- L378: "Action Accuracy (ACC), Action Accuracy (ACC)" (sic)
- Figure 4: labeling the default locations of the D classification categories would help interpret the pareto-front figure
- Figure 5: It is difficult to interpret whether to read the figure on the left, from left-to-right or right-to-left

**Questions:**

Questions:
- Q1: What is a "Hamilton" distance?  Do you mean Hamiltonian distance or Hamming distance?
  A web search produced no results for this concept.
- Q2: Wondering why the "+P_C(Y)" term is really needed, i.e., does it add value?
  The addition to the classification category number is quite strange, given that they represent different things.
- Q3: how do FID_tr and FID_te differ?
- Q4: It is still unclear how the variable-length motions are encoded and sampled.
- Q5: It is not clear how the generative model is conditioned on the previously-generated samples.
  Does this happen implicitly via the memory/GRU? If so, how is the GRU update trained so as to be incentivized
  to make the next sample highly-diverse with respect to previous samples?  This reader is very confused on this point.

---

> ### Author Response · Authors · 2024-11-20
> **Rebuttal to Reviewer MnFJ (Part 1)**
>
> *Due to the large amount of content in the reply, we have divided our rebuttal into two parts. The following is the first part.*
>
> **W1, Further Explanation on the Diversity Keeping:**
> Regarding the principle underlying our multi-objective approach for diversity maintenance, please refer to the overall rebuttal. To enhance clarity, we have included an additional description in the appendix of the revised paper (line 805 on Page 15), providing illustrative figures and diagrams designed to aid understanding, particularly about the multi-objective optimization process.
>
> **W2, Details about the Multi-Class Classifier:**
> As you noted, the multi-class classifier is indeed a key component of our method. Since the classifier is not the main contribution of this paper, we overlook the details of the multi-class classifier. In the revised paper, we have updated the description in the appendix (line 912 on Page 17).
>
> **W3, Video Results:**
> In fact, we have already presented the video results on an anonymous webpage and included a link in the abstract, though it may not have been immediately noticeable. In the revised version, we have highlighted this link in the abstract to improve visibility. Please refer to the abstract of the revised paper for this update.
>
> **W4 and Minor Comments:**
> We greatly appreciate your thorough review and detailed error corrections. Below, we address your questions item by item. Additionally, all minor issues highlighted in your review have been revised in the updated manuscript following your suggestions.
> Specifically, for the illustration of Pareto dominance, we have added a related figure in the appendix (line 820 on Page 16) using the formulated multi-objective optimization problem as an example. As for the *Smoothness Component* in the designed multiobjective function, this design is motivated by the discontinuity between different and complex motion sequences that exist in many cases [1], the smoothness component is employed to avoid this phenomenon, thereby enhancing the quality of the generated samples.
> The equation should read as $\beta(Y) = ||X_1[-1]-Y[0]|| + ||Y[-1]-X_2[0]||$ and we updated the paper accordingly. For other typos, we carefully addressed each one based on your suggestions. However, concerning the structure of the paper, we have retained it as is. This decision was made because the primary contribution of this work lies in our multiobjective modeling approach and the corresponding evolutionary multiobjective generation framework. Also, the suggested papers are also cited. In addition, we have replaced the ''batch diversity'' with ''intra-batch diversity''.
>
> [1]. Cohan S, Tevet G, Reda D, et al. Flexible motion in-betweening with diffusion models[C]//ACM SIGGRAPH 2024 Conference Papers. 2024: 1-9.

---

> > ### Author Response · Authors · 2024-11-20
> > **Rebuttal to Reviewer MnFJ (Part 2)**
> >
> > *Due to the large amount of content in the reply, we have divided our rebuttal into two parts. The following is the second part.*
> >
> > **Q1, Reply on Error Correction**: We apologize for the confusion, the correct term is indeed the Manhattan distance, or $L_1$ distance. We have updated the description accordingly to rectify this error. Please review the modified version for accuracy and clarity.
> >
> > **Q2, Effectiveness of the $P_{c}(Y)$ Term in $\alpha(Y)$**:
> > In our method, we use classifiers to predict motion patterns for each sequence, allowing us to evaluate diversity based on these differences. Specifically, $C(Y)$ indicates the category of each sequence; if two sequences are predicted to belong to different categories, we consider them to have significant differences. However, in cases where two sequences are predicted as belonging to the same category, it becomes challenging to discern their differences based solely on $C(Y)$. Therefore, we also use $P_{c}(Y)$ to further assess differences within the same category. In fact, we have already investigated the potential impact of $P_{c}(Y)$ in Section 6.2 (i.e., ablation studies on the intra-class difference), and the experimental results demonstrate that incorporating $P_{c}(Y)$ can indeed enhance our method’s performance.
> >
> > **Q3, The difference between the $FID_{tr}$ and $FID_{te}$**.
> > In the experiment, we test our method on both training dataset $FID_{tr}$ and test dataset $FID_{te}$ using the Frechet Inception Distance (FID) metric, the reason is as:
> > a. Evaluating Model Generalization: Calculating FID on the training dataset helps to understand the model's performance, while calculating it on the test dataset assesses the model's ability to generalize to unseen data.
> > b. Detecting Overfitting: If the model shows a significantly lower FID on the training set compared to the test set, it may indicate that the model is overfitting, performing well only on the training data while struggling with new data.
> >
> > **Q4, Variable-length motion generation.**
> > The proposed method supports variable-length motion generation.
> > In the sampling and encoding process of the experiment, we first calculate the cosine similarity between two given motion poses $X_1 [-1]$ and $X_2 [0]$. $X_1 [-1]$ is the last pose of sequence $X_1$, $X_2 [0]$ is the first pose of $X_2$. The desired length of the human motion sequence is set as:
> > \begin{align*}
> >     Y_{len}= Y_{min} + \left \lfloor(Y_{max}- Y_{min}) \times (1-S) \right \rfloor.
> > \end{align*}
> > where $Y_{min}$ and $Y_{max}$ are the preset minimum and maximum sequence length, respectively. In the generation of the offspring, the $Y_{len}$ is encoded into the pretrained generative model and has to be consistent with the training process details. So we introduce a padding operation to the $Y_{len}$, specifically, we repeat the last pose of the $Y_{len}$ until it achieves the $Y_{max}$.
> >
> > **Q5: Conditional generation on the previously-generated samples.**
> > We guess the reviewer is confused about the mutation operation for the generation of the offspring sequence.
> > In our method, we use the sampling method of the original VAE when generating the initial sequence.
> > In the offspring populations, the initial sequence is encoded into latent space using the pretrained Encoder, the designed mutation operation is then operated on the encoded latent space by adding noise. The noised latent space combined with the given sequence $X_1$ and $X_2$ is decoded into the offspring sequence using the pretrained Generator. The generation formula is described in line 256 on Page 5 in equation (5) of the original paper.
> > The formulation of the mutation process (i.e., the exploration process for motion sequence generation) is as:
> > \begin{align}
> >     \begin{aligned}
> >     & Z  \leftarrow  Encoder(Y_{[i-1]}^n) \\
> >     \end{aligned}
> > \end{align}
> > \begin{align}
> >     \begin{aligned}
> >     & Z  \leftarrow  Z + noise  \\
> >     \end{aligned}
> > \end{align}
> >
> > \begin{align}
> >     \begin{aligned}
> >     & Y_{[i]}^n  \leftarrow  Generator(Z)
> >     \end{aligned}
> > \end{align}

---

> ### Author Response · Authors · 2024-11-24
> **Look forward to hearing your thoughts**
>
> Dear Reviewer,
> We hope the explanations and responses we provided have addressed your concerns. If there are any remaining questions or areas of confusion, we would be more than happy to continue the discussion and provide further clarification. Your feedback is invaluable to us, and we look forward to hearing your thoughts.

---

> ### Comment · Reviewer_MnFJ · 2024-11-24
> **Reply to Rebuttal comments**
>
> Thank you for the detailed rebuttal comments and the related paper edits.
>
> - re: change of "Hamilton" to "Manhattan" distance -- thank you, that makes sense. Note: there are still two remaining mentions of "Hamilton" distance in the paper
> - re: additional figures illustrating the pareto optimal front, now in Appendix -- thank you, this is helpful.  I suggest further providing the category labels in Figure 6 and 7, i.e., for Fig 7(a) and considering beta=0, label these points as categories 1,2,3,4,5.  Conceptually, however, it is strange to see that different degrees of "interpolation smoothness", i.e., given by the addition of the $$\beta(Y)$$ term, can create overlap with adjacent class categories. This is visible to some degree in Fig 6. Equivalently, in Eqn 4, the terms for F1 and F2 are mixing quantities that represent different quantities, with different units. Shouldn't there be at least be a scalar weighting factor somewhere to accommodate this?  Wouldn't it be simpler to have a simple criterion of "best smoothness" within each action category, as opposed to the artificial combination of these two different criteria, i.e., (a) discrete motion category, and (b) smoothness? Are all motion categories always applicable to all motion in-betweening problem instances?
> - re: videos -- thank you, I now see the GitHub page.  It would be worthwhile explicitly mentioning the availability of video results in the paper. The quality of the video results is disappointing, however, in three respects: (i) the stick-figure motion representation; (ii) the lack of visible references on the ground plane to help the viewer better judge the overall character movement with respect to the ground and the quality of the motion contacts made with respect to the ground; and (iii) the overall motion quality (speaking qualitatively, irrespective of FID scores).  Using the skinned SMPL meshes, as used for the static figures, would help the readers better judge the motion quality for themselves.
> - re: OmniControl and CondMDI (as referred to in other reviews) -- refering to the need for hyperparameter optimization as the principal critique of these two works is an unsupported argument, in the absence of experiments (or other documentation) that compares the need for (or sensitivity to)  hyperparameter optimization across these methods as well as the proposed method.
>
> Overall, I greatly appreciate the revisions and explanations, which have improved my understanding. However, I am keeping my overall score unchanged, given (a) remaining confusion (for this reader) as to why it is meaningful to combine motion-category and motion smoothness into a single space for pareto-optimization; (b) limited ability to judge motion quality, as compared to current SOTA methods, commonly rendered with skinned meshes, a full ground plane, etc.

---

> > ### Author Response · Authors · 2024-11-25
> > **Re-Reply to Rebuttal Comments (Part 1)**
> >
> > **Re1**:
> > Thanks for you comment. We have corrected these typos.
> >
> > **Re2**:
> > 1. Further explanation of equation (4)}: Thank you for your reply. It seems that some details remain unclear. Additionally, I guess there appears to be a misunderstanding regarding the content presented in Figures 6 and 7. Specifically, Figure 6 illustrates the influence of $\beta(Y)$, whereas Figure 7 primarily addresses the influence of $\alpha_1(Y)$ and $\alpha_2(Y)$. It is important to note that the result in Figure 7(b) is not caused by $\beta(Y)$; rather, the lack of diversity (in motion categories) leads to the situation observed in Figure 7(b), even when smoothness is satisfied (i.e., $\beta(Y)=0$). To clarify these points further, we provide a further explanation based on three scenarios shown in Figure 6, Figure 7(a), and Figure 7(b).
> >
> > (1) **Figure 6**: The motion sequences that cannot guarantee smoothness (e.g., $Y_c$ and $Y_d$ in Figure 6) will be Pareto dominated by motion sequences that can guarantee smoothness (e.g., $Y_a$ is better than $Y_c$ on both of the two objectives, and $Y_b$ Pareto dominates $Y_d$ also). Therefore, we need to find the Pareto optimal solutions of equation (4) to ensure the smoothness of the motion sequences.
> >
> > (2) **Figure 7(b)**: Even if all sequences can ensure smoothness, if their diversity—manifested in the diversity of motion categories—is not strong enough, they will lose diversity in the \textit{objective space}.
> >
> > (3) **Figure 7(a)**: The motion sequence batch that ensures Pareto optimality and diversity in the \textit{objective space} is the desired outcome. This goal can be achieved through evolutionary algorithms, as these methods are highly adept at this type of task.
> >
> > The key insight from these scenarios is that \textit{the diversity in the objective space of equation (4) can be translated into the intra-batch diversity of the motion sequences}, and this goal can be easily achieved using evolutionary algorithms or related techniques. Moreover, we also have updated Figures 6 and 7 to further enhance the understanding.
> >
> > 2. About whether it is necessary to add a weight for $\beta$: In the context of our problem, specifically in $F_1$ and $F_2$, it is unnecessary to introduce a weight for $\beta$. Below, we provide an explanation of why this is feasible. In conventional machine learning scenarios, where multiple objectives or loss functions are involved, it is common practice to weight and aggregate them. This stems from the fact that such objectives or losses often conflict with each other—minimizing one may lead to the deterioration of others. Achieving simultaneous optimization of all losses in these cases is inherently challenging. The primary purpose of introducing weights in these scenarios is to balance the trade-offs among different losses, thereby ensuring a more equitable reduction across all objectives. However, our situation is different. In our case, finding smooth sequences to achieve $\beta \approx 0$ is relatively straightforward, and many motion sequence categories can satisfy this condition. There is no clear conflict between $\alpha_1$ (or $\alpha_2$) and $\beta$. Even if a conflict arises in certain categories (i.e., categories may not be applicable to some motions under some conditions, as explained below), it is unlikely to occur across all categories. Thus, scalar weights are unnecessary in this context.
> >
> > 3. Regarding your suggestion to establish a straightforward criterion of 'best smoothness' within each motion category, we must admit that we are unclear about the intent of your question. Are you suggesting that we define a multiobjective optimization problem with two objectives: one objective is the category predicted by the classifier and another is the smoothness indicator? If so, we believe this is not an ideal approach, as these two objectives are not in conflict (as discussed previously with $\alpha_1$ and $\beta$), and this setup would not align with the requirements of multiobjective optimization. In fact, this is also the rationale behind setting $\alpha_2 = 1 - \alpha_1$ in equation (2), as this configuration ensures that $F_1$ and $F_2$ are always in conflict with each other.
> >
> > 4. Regarding the question of whether all motion categories are always applicable to all in-between problem instances: if certain categories are not applicable, the corresponding Pareto optimal solution may not exist or may be challenging to obtain. This would manifest in the objective space as a discontinuity in the Pareto front. Evolutionary algorithms can adaptively handle this issue, which can be referred to in the paper we cited on line 251 on page 5 [1] or related to SOTA multiobjective evolutionary algorithms.
> >
> > [1] Deb, K., Pratap, A., Agarwal, S., \& Meyarivan, T. A. M. T. (2002). A fast and elitist multiobjective genetic algorithm: NSGA-II. IEEE transactions on evolutionary computation, 6(2), 182-197.

---

> > ### Author Response · Authors · 2024-11-25
> > **Re-Reply to Rebuttal Comments (Part 2)**
> >
> > **Re3**:
> > Our method is based on the pretrained backbone generative model, the quality and ground plane problem are alleviated by the generative model, and the proposed module of our method will not influence the quality of generated human motions. In contrast, the proposed method can further enhance the diversity based on the backbone generative model. The skinned meshes videos can be rendered through the rotation-based representation. Due to the large volume of data, it will take some time to process. If needed, we can provide the results later.
> >
> > **Re4**: Thanks for your comment. Following your suggestion, we have refined and updated the summary of the related work. Now, we have modified this part, which is focusing on the sampling with intra-batch diversity, thus aligning with our topic. Please see line 147 on page 3 for details.

---

> > ### Author Response · Authors · 2024-11-27
> > **Re-Reply to Rebuttal Comments (Additional Videos)**
> >
> > Dear Reviewer, according to your concern, we have added additional videos results in skinned meshes style, which can be found on the anonymous Github mentioned in the abstract. Note that the motion skeleton data representation generated by the designed model needs to be rendered into human mesh format using Blender. This process is time-consuming and the quality of the generated visuals will influenced by the renderer itself. Moreover, we would like to clarify that, the generated in-betweening motion sequences by our method can achieve the transitions between complex action cues regardless of the data format used for presentation, rather than generating in-betweening sequences based on the text and sparse data frames referred in the papers by the reviewer (although our method has the potential to be extended to these tasks and the multiobjective generative models for intra-batch diversity is a promising direction, as mentioned in our responses for reviewer jtPA).
> >
> > Additionally, a more detailed explanation of the principle has been provided in the previous rebuttal. We hope these additions will further enhance your understanding of our methods. Engaging in further discussions regarding our approach would be a pleasure, and we look forward to hearing your ideas.

---

### Official Review · Reviewer_kZ98 · 2024-11-03

**Soundness:** 3
**Presentation:** 3
**Contribution:** 2
**Rating:** 5
**Confidence:** 2

**Summary:**

Authors propose a multi-objective framework for the human motion in-betweening task. More specifically, they introduce a bi-objective optimization problem; optimizing the diversity and smoothness of transitions between keyframes and generated motions. The proposed optimization framework is applied on top of a pretrained generative model to produce the final diverse completed motions.

**Strengths:**

* The problem of diversity in human motion generation/completion is an important problem. One of the major limitations of most generative models developed for the in-betweening task, is the lack of diversity due to overfitting.
* Paper is well-structured and easy to read.
* The experiment section contains most of the important in-betweening baselines.

**Weaknesses:**

* My main concern with this work is the substance of the paper. To me, it primarily appears that the proposed framework suggests iteratively sampling from a pretrained generative model, rejecting samples that lack sufficient smoothness while ensuring diversity among the generated samples. The only substantial contribution seems to be the definition of a bi-objective function to balance both diversity and smoothness. However, I am not entirely convinced that this specific framing is necessary; simply rejecting samples based on smoothness and applying a diversity metric across the entire sample set might have been equally effective.

* While I appreciate the authors' thorough investigation of related work and relevant baselines, some recent in-betweening studies are missing from the discussion and experiments sections. For example, CondMDI (Cohan et al., 2024) and OmniControl (Xie et al., 2023) are notable recent works that should be considered.

* The effectiveness of the proposed method relies heavily on the performance of the underlying generative model. If the backbone generative model has limited diversity, the iterative framework is unlikely to significantly enhance diversity.

**Questions:**

1. I am having trouble understanding the diversity component as defined in Equation 2. Isn't $C(Y)$ a single value between $0$ and $D-1$ and $P_c(Y)$ the probability of Y belonging to class $c$? Can you elaborate on how this definition ensures diversity?

2. Can you explain how your model differs from simple rejection sampling, where $N$ motion sequences are iteratively generated and the $M<N$ sequences with the highest diversity (using any heuristic or method, such as pairwise distance maximization) are retained? Then, samples below a certain smoothness threshold are rejected, and this cycle repeats until $N$ samples are obtained.

3. I believe it would be beneficial to compare the proposed method with straightforward rejection-based sampling approaches to highlight the unique advantages of your method.

4. I highly recommend that the authors include recent relevant diffusion works of CondMDI (Cohan et al., 2024) and OmniControl (Xie et al., 2023) in the related work section (Section 2.2).

5. As with any iterative or rejection-based method, inference time is a key consideration. What is the inference time with this method? How does it compare with SOTA methods?

6. What is the data representation used here? Is it global joint positions? How are the keyframes defined?

7. I am curious to know the details of the backbone generative models used in this work. Particularly, how they condition on keyframes. I think these details need to be added to the supplementary material.

---

> ### Author Response · Authors · 2024-11-20
> **Rebuttal to Reviewer kZ98 (Part 1)**
>
> *Due to the large amount of content in the reply, we have divided our rebuttal into two parts. The following is the first part.*
>
> **Q1, The principle of the constructed multi-objective optimization problem:**
> Your understanding is correct: $C(Y)$ represents a value between $0$ and $D-1$, $P_{c}(Y)$ denotes the probability of $Y$ belonging to class $c$. Regarding the principle underlying our multi-objective approach, we have provided a detailed explanation in the Overall Rebuttal. Additionally, to clarify why we say that the diversity component ensures diversity: as outlined in our overall rebuttal, we do not rely on any indicator function to directly define diversity. Instead, the diversity component's role is to ensure that the two objective functions in the multi-objective problem conflict with each other—that is, a low value in one objective corresponds to a high value in the other. This conflict is a fundamental characteristic of the multi-objective optimization problem and guarantees the existence of a Pareto front, making it meaningful to maintain diversity in the objective space, while allowing for the use of evolutionary algorithm techniques (specifically in the *environmental selection* step of our algorithm) to approximate it with a diverse set of sequences. In summary, the diversity component ensures the existence of a Pareto front, while evolutionary algorithm techniques facilitate the acquisition of diverse sequences along this front.
>
> **W1, Q2, and Q3, Difference and Comparison with Rejection Sampling Method:** Thank you for your suggestions. As you indicated, distinguishing between the proposed method and the rejection-sampling method is essential. Here, we clarify their differences from an optimization perspective. We posit that the rejection-sampling method can be regarded as a heuristic optimization algorithm designed for the following single-objective optimization problem: $\min: -F_{div}(\mathcal{Y}) + F_{smooth}(\mathcal{Y})$, where $\mathcal{Y} = ( Y_1, \ldots, Y_N )$ represent a set of $N$ in-betweening motions, $F_{div}$ and $F_{smooth}$ measures the diversity and the motion smoothness of $\mathcal{Y}$, respectively. The rejection sampling method introduced by you can be considered as a heuristic optimization approach for this single-objective optimization with the following structure: it first adjusts a subset of in-betweening motions in $\mathcal{Y}$ to maximize $F_{div}(\mathcal{Y})$ (i.e., generating $N$ motion sequences and retaining the $M < N$ sequences with the highest diversity). It then adjusts another subset of in-betweening motions in $\mathcal{Y}$ to minimize $F_{smooth}(\mathcal{Y})$ (i.e., rejecting samples that fall below a specified smoothness threshold). By alternately optimizing $F_{div}(\mathcal{Y})$ and $F_{smooth}(\mathcal{Y})$ heuristically, rejection sampling can generate diverse motions. However, $F_{div}(\mathcal{Y})$ and $F_{smooth}(\mathcal{Y})$ may have conflicting optimization objectives, meaning that increasing diversity may reduce smoothness, and vice versa. This conflict can lead to an oscillatory optimization process, slowing convergence. In contrast, the proposed multiobjective optimization based approach fully considers the trade-off between diversity and smoothness, resulting in a more efficient optimization process. In addition, we have also conducted an experiment to compare the results of rejection sampling with those of our proposed method, please see the overall rebuttal for all authors. The results demonstrate that our method achieves superior diversity.
>
> **W2, Q4, Investigation of related work on CondMDI and OmniControl:**
> Thank you for your valuable feedback. We overlooked the recent relevant diffusion works, ''CondMDI" (Cohan et al., 2024) and ''OmniControl" (Xie et al., 2023). As suggested, we have included them in the revised manuscript (Section Related Work, line 140 on Page 3).

---

> ### Author Response · Authors · 2024-11-20
> **Rebuttal to Reviewer kZ98 (Part 2)**
>
> *Due to the large amount of content in the reply, we have divided our rebuttal into two parts. The following is the second part.*
>
> **W3, Potential Influence of the Backbone Generative Model:**
> As you mentioned, the performance of our method does depend on the performance of the backbone model. However, as discussed in our overall rebuttal about the advantages, if the backbone model has strong diversity capabilities, our method can also be embedded into it to further enhance the diversity of batch sequences. In fact, even relatively weaker backbone models can achieve competitive results with state-of-the-art generative models when integrating our method, as demonstrated results in our experiment. We believe that such a plug-and-play framework remains highly valuable for a wide range of applications.
>
> **Q5, The Inference Time Consumption of the Proposed Method:**
> Thank you for your comment. Regarding the inference time of our method, we have addressed this point in the overall rebuttal for all reviewers. Please refer to the overall rebuttal for a detailed discussion. It is worth noting that the rejection sampling in our experiment follows the same number of iterations as the proposed method. However, as per the reviewer’s suggestion, each iteration of rejection sampling involves two steps. This could explain why rejection sampling consumes more time compared to our method.
>
> **Q6, Q7, Human data representation and keyframes:**
> The data representations for human motions are diverse, such as rotation-based and position-based representations. In this work, we use a rotation-based representation for human motions. Position-based representations may encounter issues with unrealistic bone lengths due to prediction inaccuracies; Rotation-based representation for each joint can preserve a reasonable human bone length ratio. Therefore, we employ the rotation-based representations.
>
> The used keyframes in this paper are different from the CondMDI and OmniControl settings. These two papers are conditioned by a given text prompt and spatial control signals. The keyframes are sparsely placed in every *T* frame, which is implemented by the DDPM.
> In this work, our condition is only with the user-provided complex keyframes without text prompt. The length of the keyframes can be variable in our method and set to 5 in the experiment. The focus of our paper is the diverse and accurate in-betweening motion sequences with the designed ''intra-batch diversity'', which is not sensitive to the hyperparameter and can achieve the balance between diversity and accuracy. As the reviewer suggested, we have included the details in the experimental setting (line 404 on Page 8).

---

> ### Author Response · Authors · 2024-11-24
> **Look forward to hearing your thoughts**
>
> Dear Reviewer,
> We hope the explanations and responses we provided have addressed your concerns. If there are any remaining questions or areas of confusion, we would be more than happy to continue the discussion and provide further clarification. Your feedback is invaluable to us, and we look forward to hearing your thoughts.

---

> ### Comment · Area_Chair_RWnx · 2024-11-25
> **Please read rebuttal**
>
> Dear Reviewer kZ98,
> Dear Reviewer zDKg, Could you please read the authors' rebuttal and give them feedback at your earliest convenience? Thanks.
> AC

---

### Author Response · Authors · 2024-11-20
**Overall Rebuttal**

We greatly appreciate the suggestions and comments provided by the reviewers for this paper. The comments provided by the reviewers are of great reference value. Here, we reply to the concerns of most reviewers and provide explanations and answers. In addition to these summarized opinions, we have also provided separate responses to each reviewer's comments.

**The Principle Behind the Multi-Objective Modeling**:
We notice that, the reviewers have raised concerns regarding the diversity maintenance principles in the multi-objective modeling process in the proposed method. Here we further clarify the underlying principles of our approach. Our proposed multi-objective optimization-based approach diverges from the intuitive idea of using a diversity metric for the whole batch (e.g., the average pairwise distance) to evaluate and enhance the diversity by optimizing the diversity metric. Instead, the diversity-preserving capability is driven jointly by the structure of the multi-objective optimization problem itself and the techniques inspired by evolutionary algorithms. First, we show that, by uniformly distributing objective function values corresponding to the sequences along the Pareto front, our approach ensures a diverse set of motion sequences, as supported by Theorems 1 and 2. Second, multi-objective evolutionary algorithms enable efficient exploration of such sequences. In summary, the formulation of the multi-objective optimization problem defines the properties of diverse and smooth batch sequences, while evolutionary algorithm techniques facilitate their effective identification.

**Concern on the Time Consumption**:
Some reviewers have expressed concerns regarding the sampling time of the proposed method.
We conduct ablation studies to support our choice. As shown in Table. 1,  we compare our method with various approaches based on sampling time, as well as the corresponding FID and APD metrics.
It can be observed that, as an iterative method, our approach generates more diverse and smooth sequences within the same number of iterations compared to similar methods, such as rejection sampling. This means that, for sequences of comparable quality, our method requires less sampling time than existing iterative methods of the same type.
Furthermore, with only a few iterations, our method achieves performance competitive with current state-of-the-art DDPM-based motion generation models without any additional training process, this is obviously very cost-effective.
It is noted that there are several algorithms designed to accelerate the sampling process of DDPM, such as DDIM and Flow Matching. However, these methods often suffer from low generation performance or lack diversity due to the nature of ordinary differential equations. Our approach effectively addresses these drawbacks through a specially designed multi-objective function. This function enhances the diversity and quality of generated motions without significantly increasing sampling time. Additionally, the multi-objective function can be adapted for different tasks, making our method highly feasible for practical applications.

| Method                  | FID_{tr} | FID_{te} | APD  | Sampling Time(s) |
|-------------------------|------------|------------|------|------------------|
| DDPM                    | 20.02      | 19.81      | 8.10 | 9.03             |
| DDIM                    | 21.54      | 21.39      | 8.24 | 7.14             |
| MGF-IMM(DDIM+Rejection) | 15.52      | 15.24      | 8.29 | 11.63            |
| MGF-IMM(DDPM+EC)        | 14.32      | 13.65      | 8.45 | 10.74            |
| MGF-IMM(DDIM+EC)        | 14.33      | 13.63      | 8.47 | 9.10             |

**Further Discussion on the Advantage of the Proposed Approach**:
In addition to the discussion above, we would like to further emphasize the advantages of the proposed method. In our approach, the core mechanism for enhancing diversity lies in the multi-objective modeling and the multi-objective generation framework inspired by evolutionary algorithms. While we present a model structure specifically for in-betweening motion generation in this paper, our method is not limited to any specific generation model or network architecture. This flexibility means that as more advanced generative models or methods emerge, our multi-objective modeling approach and multi-objective generation framework can be seamlessly embedded into these new models, providing an opportunity to further enhance batch generation diversity. This is also why we say "Unlocking" in our title. We believe that such a plug-and-play framework is highly valuable and straightforward to implement, making it practical for a wide range of applications.

---

### Author Response · Authors · 2024-11-22
**Official Comment by Authors**

Dear Reviewers,

We sincerely thank you for your precious time and efforts in reviewing our paper.

We would like to inquire whether our response has addressed your questions and concerns. We are more than happy to discuss with you further and provide additional materials.

Best regards,

The Authors

---

### Meta-Review · Area_Chair_RWnx · 2024-12-17

**Metareview:**

The authors propose a multi-objective motion in-betweening technique. The approach is on the basis of existing generative models. However, the clarity of the paper needs improvement. While some reviewers criticized the use of pre-triained model, I disagree with the argument as it is common practice in the era of big models. My biggest concern is about the video quality. The results seem unsatisfactory when it is animated. For example, the walk and throw example has an obvious pause and motion leap. Moreover, the manuscript missed one or a few references.

**Additional Comments On Reviewer Discussion:**

Most of the reviewers' concerns are not fully addressed.

---

### Decision · Program_Chairs · 2025-01-22

Reject